# The Effect of Plant Additives on the Stability of Polyphenols in Cloudy and Clarified Juices from Black Chokeberry (*Aronia melanocarpa*)

**DOI:** 10.3390/antiox9090801

**Published:** 2020-08-27

**Authors:** Andrzej Sidor, Agnieszka Drożdżyńska, Anna Brzozowska, Artur Szwengiel, Anna Gramza-Michałowska

**Affiliations:** 1Department of Gastronomy Sciences and Functional Foods, Faculty of Food Science and Nutrition, Poznań University of Life Sciences, Wojska Polskiego 31, 60624 Poznań, Poland; andrzej.sidor@up.poznan.pl (A.S.); anna.brzozowska@up.poznan.pl (A.B.); 2Department of Biotechnology and Food Microbiology, Faculty of Food Science and Nutrition, Poznań University of Life Sciences, Wojska Polskiego 31, 60624 Poznań, Poland; agnieszka.drozdzynska@up.poznan.pl; 3Institute of Food Technology of Plant Origin, Faculty of Food Science and Nutrition, Poznań University of Life Sciences, Wojska Polskiego 31, 60624 Poznań, Poland; artur.szwengiel@up.poznan.pl

**Keywords:** black chokeberry, *Aronia melanocarpa*, cinnamon, *Cinnamomum*, clove, *Syzygium aromaticum*, juice, polyphenols, anthocyanin

## Abstract

Black chokeberry (*Aronia melanocarpa*) is a fruit with increasing popularity in consumption and processing. Recent research has strengthened the position of chokeberry as a source of phenolic compounds, antioxidants with high pro-health values, therefore it is important to investigate other substances protecting biologically active compounds during juice processing. This study was an attempt to reduce the loss of polyphenol in cloudy and clarified chokeberry juice by adding aqueous cinnamon and clove extracts. The results showed that the clarification of juices did not cause significant changes in the concentration of polyphenols. However, the addition of plant extracts prior to pasteurisation process influenced the content of phenolic compounds in the chokeberry juices. The main change in the composition of the chokeberry juices observed during storage was a result of the degradation process of anthocyanins. The research showed that, despite the common view about the beneficial effects of polyphenols and other compounds exhibiting mutual antioxidative potential, it is very difficult to inhibit the degradation process.

## 1. Introduction

Black chokeberry (*Aronia melanocarpa*, *Rosaceae*) is a perennial fruiting shrub that is gaining popularity [1,2]. The plant has various advantages: its fruits have great health-promoting potential, it is relatively easy to cultivate, and it yields large amounts of fruit [3,4]. The properties of chokeberry are mostly related with its high antioxidative activity that results from the presence of polyphenols, including cyanidin-3-*O*-glycosides (galactoside, arabinoside, xyloside and glucoside), quercetin (galactoside, glucoside, rutinoside, vicianoside and robinobioside), chlorogenic and neochlorogenic acids, and proanthocyanidins [5,6,7,8,9,10,11,12].

The availability of fresh fruits is periodic (from mid-August to late September), and the short shelf-life necessitates their further processing. Chokeberries are processed into juices, dried fruits, jams, syrups, teas, tinctures, jellies, food colourings, dietary supplements, etc. Fruit processing involves changes in the content of bioactive compounds resulting from the effect of chemical and physical factors [13]. During processing, fruit components are degraded mostly by increased temperature, oxygen and enzymes, e.g., polyphenol oxidase (PPO) [14]. The results of many studies do not support the fact that the addition of antioxidants from other sources prevents loss of native compounds. In fact, the unpredictable effect of enhancing or suppressing activity in a mixture of different active compounds is often overlooked. Therefore, researchers are conducting investigations to understand the effects and effectiveness of interactions between raw materials, pure compounds and products with antioxidative properties. Furthermore, the addition of plant extracts could increase the biological potential of the obtained drinks. Cinnamon has antitumor, antiinflammatory, antidiabetic, antiobesity, antibacterial, antiviral, cardiovascular protective, cytoprotective, neuroprotective and immunoregulation activity [15]. Cloves pharmaceutical purposes are antimicrobial (antibacterial, antifungal, anticandidal), antiviral, antiinflammatory and anticancer [16,17]. Both spices, cinnamon and cloves contain polyphenolic compounds. Cinnamon contains phenolic acids (caffeic, chlorogenic, ferulic, *p*-coumaric, *p*-hydroxybenzoic, protocatechuic, rosmarinic, syringic), flavonols (rutin, quercetin, quercitrin, kaempferol, isorhamentin), flavone (apigenin), flavanols (catechin, epicatechin), procyanidins, and non-flavonoid phenoglycosides [18,19,20,21]. In contrast, cloves contain phenolic acids (gallic, caffeic), flavonols (tamarixetin 3-O-β-D-glucopyranoside, ombuin 3-O-β-D-glucopyranoside, quercetin) and ellagitannins (casuarictin, eugeniin, tellimagrandin I and 1,3-di-Ogalloyl-4,6-O-(S) -hexahydroxydiphenoyl-β-D-glucose [22,23,24].

In this study, the profile of bioactive compounds in chokeberry juices enriched with cinnamon or clove extracts subjected to processing has been analysed for the first time. Therefore, the aim of the study was to analyse the influence of spices extracts addition, the technological processes (clarifying, pasteurisation) and the storage time on the changes of selected polyphenols content in chokeberry juices.

## 2. Materials and Methods

### 2.1. Reagents

The following HPLC- or LC-MS-grade reagents were used for analyses: acetonitrile, L-ascorbic acid, methanol (Sigma-Aldrich, Poland), acetic acid (Baker, Poland), and formic acid (VWR Int., Poland). The qualitative and quantitative content of polyphenolic compounds was determined with the following standards: 3,4-dihydroxybenzoic acid, 4-hydroxybenzoic acid, chlorogenic acid, caffeic acid, gallic acid, isovanilic acid, kaempferol, luteolin, neochlorogenic acid, orientin (luteolin-8-glucoside), *p*-coumaric acid, procyanidin B2, quercetin, quercetin-3-*O*-glucoside (Sigma-Aldrich, Poland), (+)-catechin, (–)-epicatechin, quercetin-3-*O*-galactoside, quercetin-3-*O*-rutinoside, quercetin-3-*O*-vicianoside (Extrasynthese, France), cyanidin-3-*O*-arabinoside, cyanidin-3-*O*-galactoside, cyanidin-3-*O*-glucoside (Polyphenols Laboratories AS, Norway) and cyanidin-3-*O*-xyloside (Toronto Research Chemicals, Canada).

### 2.2. Sample Preparation

The chokeberry fruits (*Aronia melanocarpa*) were purchased at the Agricultural and Orchard Experimental Farm (Przybroda, Poland). According to the guidelines of the Polish Standard PN-R-75032: 1996 [25], the fruits belonged to the extra quality class. Cinnamon spice plants (KOTÁNYI Polonia, Poland) and cloves (McCormick, Poland) were purchased in a chain store.

On harvesting, the fruits were cleaned, separated and washed. The clean fruits were frozen and stored for 8 weeks. Before pressing juice, the fruits were thawed at about 20 °C and processed with a blender. Chokeberry puree was pressed in a bladder press at a pressure of 3 bar for 5 min. Obtained juice was diluted with water at a 3:1 ratio and then divided into two parts. Part of the juice was clarified by membrane filtering. The efficiency of the process was visually determined by the absence of suspended high molecular solids such as protein. The other part was left cloudy. Clove and cinnamon aqueous extracts were prepared according to the method developed by Gülçin et al. [26]. The spices were ground and added to boiling water at a ratio of 1:20 and boiled for 15 min. The cooled extract was filtered. The clear and cloudy juices were poured into 120 mL glass jars with galvanised and varnished steel with a gasket closures then cinnamon or clove extract were added in proportion 95:5 *v*/*v*. Volume juice/drink in each jar was 100 mL, while the addition of extracts at the level of 5% was preceded by sensory optimisation using a consumer panel. Juices with ascorbic acid contains ascorbic acid addition to final concentration 300 mg/L according to nutrition standards [27]. Juices without additives were taken as controls. The juices were closed with lids and pasteurised at 80 °C for 10 min [28]. Next, they were cooled with a stream of cold water and stored for 90 days in a dark place at ambient temperature (20 ± 1 °C). Each juice/drink version was prepared in triplicate.

### 2.3. HPLC Analysis of Chokeberry Juices/Drinks Polyphenolic Compounds

The juices/drinks for chromatographic analysis were diluted with a mixture, which consisted of 150 mL HPLC grade methanol ≥99.9%, 345 mL deionised water, 0.5 mL HPLC grade acetic acid ≥99.7% and 1 g ascorbic acid [29]. The diluted juices/drinks were purified by passing through 0.45 µm PTFE syringe filters. Next, they were analysed.

An Agilent Technologies 1200 series liquid chromatograph (Agilent Technologies, USA) equipped with an autosampler (G1329B), pump (G1312B) and diode detector (G1315C) with a spectrum range of 190–400 nm was used for the analysis. The signal was recorded for four wavelengths corresponding to the following groups of compounds: polyphenols (280 nm), phenolic acids (320 nm), flavonols (360 nm), anthocyanins (520 nm). An SB-C18 column (50 mm × 4.6 mm, 1.8 µm particle diameter, Agilent Technologies, USA) thermostatted at 25 °C was used for measurements. The following eluents were used: A—4.5% formic acid (LC-MS grade 99%), B—acetonitrile (HPLC grade ≥99.9%) at a flow rate of 1 mL/min, gradient: 0 min 3% B, 7 min 9% B, 13 min 12% B, 20 min 14% B, 21 min 80% B, 26 min 80% B, 27 min 3% B, 36 min 3% B. 10 µm samples were applied to the column [30,31]. The ChemStation for LC 3D systems program (Agilent Technologies, USA) was used to compare spectra and retention times with standards so as to identify compounds and do quantitative calculations.

### 2.4. Analysis of Polyphenolic Compounds of Spice Plants Extracts in the LC-MS System

The analysis of polyphenolic compounds in spice plant extracts was performed using reverse phase chromatography using mass spectroscopy on a Dionex UltiMate 3000 UHPLC device (Thermo Fisher Scientific, USA) coupled with a Bruker maXis ultra-high-resolution tandem spectrometer (Bruker Daltonik, Germany) using a quadrupole and a time-of-flight analyser. Ionisation was carried out by electrospray in negative ion mode. A Kinetex ™ 1.7 µm C18 100 A, 100 × 2.1 mm column (Phenomenex, USA), thermostated at 40 °C, was used for the chromatographic separation. The mobile phase was solvent A-water with 0.1% formic acid and solvent B-acetonitrile. The flow rate was set at 0.2 mL/min with a linear elution gradient of 5 to 95% of component B over 20 min. The volume of the injected sample was 3 μL. The following mass spectrometry parameters with the ESI source were used: capillary voltage 4500 V, nitrogen nebulisation at 1.8 bar pressure, drying gas flow (N_2_) 9 L/min at 200 °C. The ion signal was collected in the range of 80–1200 m/z. The ESI-MS system was calibrated with sodium formate salt, the molecular weight standard was dosed each time at the beginning of the chromatographic separation.

The results were developed using Data Analysis 4.1 (Bruker Daltonik, Germany). Peaks corresponding to the ions of the analysed compounds [M-H] were extracted from the chromatogram and then integrated. Compounds were identified by comparing retention times with standards, and based on molecular weight and structural information from a mass spectrometer. Quantitative analysis was carried out based on prepared standard curves. Results are expressed in µg/mL of extract.

### 2.5. Anthocyanin Degradation Kinetics

Cyanidin-3-*O*-galactoside, arabinoside, xyloside and glucoside degradation over time for each treatment was plotted using first-order reaction rate kinetics and the equation:(1)$$\ln C_{t}=\ln C_{0}-kt$$
where *C_t_* is the total concentration at time *t*, *C*_0_ is the initial concentration at time zero, *k* is the first-order rate constant and time *t* is the storage time (degradation reaction time) in days. The anthocyanin half-life in each mixture was calculated using the equation:(2)$$t_{1/2}=\frac{\ln2}{k}=\;\frac{0.693}{k}\;\left[d\right]$$

### 2.6. Statistical Analysis

The results are expressed as the mean ± standard deviation. Analysis of one-way variance (ANOVA) was used to study effect of extracts or ascorbic acid solution addition on polyphenols stability after addition to juice, next during clarification, pasteurisation and storage. Differences between samples were assessed by Tukey’s HSD post hoc test. Differences were considered statistically significant when *p* ≤ 0.05. Software used for ANOVA test was Statistica 13 (StatSoft, Poland). Reaction rates for anthocyanin degradation were obtained from linear regression conducted using Microsoft Excel 2016.

## 3. Results and Discussion

### 3.1. Profile of Polyphenolic Compounds in Juices/Drinks

Chokeberry juices are rich in polyphenols. The main polyphenols of chokeberry fruits and products are: phenolic acids (chlorogenic and neochlorogenic), flavonols (quercetin-3-*O*-glucoside, galactoside, rutinoside, vicianoside and robinobioside), anthocyanins (cyanidin-3-*O*-galactoside, arabinoside, glucoside and xyloside) and proanthocyanidins [2]. Determination of the polyphenol content in the aqueous cinnamon extract showed the presence of: 3,4-dihydroxybenzoic acid (6.16 ± 0.37 µg/mL), chlorogenic acid (1.16 ± 0.00 µg/mL), *p*-coumaric acid (0.07 ± 0.00 µg/mL), catechin (5.10 ± 0.11 µg/mL) and procyanidin B2 (2.37 ± 0.21 µg/mL). The aqueous extract of cloves had more diverse profile of polyphenols: 3,4-dihydroxybenzoic acid (0.40 ± 0.04 µg/mL), 4-hydroxybenzoic acid (0.34 ± 0.02 µg/mL), caffeic acid (1.00 ± 0.03 µg/mL), chlorogenic acid (48.41 ± 3.00 µg/mL), gallic acid (18.7 ± 0.81 µg/mL), isovanilic acid (0.31 ± 0.01 µg/mL), *p*-coumaric acid (0.32 ± 0.02 µg/mL), kaempferol (0.31 ± 0.01 µg/mL), luteolin (0.12 ± 0.00 µg/mL), orientin (0.22 ± 0.02 µg/mL), quercetin (0.57 ± 0.01 µg/mL), quercetin 3-D-galactoside (0.45 ± 0.03 µg/mL).

The analysis of the composition of the juices/drinks revealed the presence of basic chokeberry polyphenolic compounds. The content of neochlorogenic and chlorogenic acids in the unpasteurised juices/drinks amounted to 30.03–34.77 mg/100 mL and 27.28–32.13 mg/100 mL, respectively (Table 1). Kardum et al. [32] found similar content of phenolic acids in chokeberry juices: 28 mg/100 g (neochlorogenic acid) and 32 mg/100 g (chlorogenic acid). Mayer-Miebach et al. [33] found 21–29 mg/100 mL (neochlorogenic acid) and 20–30 mg/100 g (chlorogenic acid) in chokeberry juice. There were small amounts of flavanols in the unpasteurised juices. The juices contained very little catechin and (−)-epicatechin, regardless of the additive used, in relation to the other compounds found in the tested juices / drinks. It is noteworthy that the content of (−)-epicatechin was more than twice as high as the content of (+)-catechin (Table 1). The content of (+)-catechin in chokeberry fruit and products is so low that this compound is usually omitted in analyses [34,35,36]. It should be noted that their evaluation is advisable, because technological processes can affect the decomposition of proanthocyanidins, also known as condensed tannins, the building blocks of which include catechin and epicatechin [37].

The extracts of cinnamon, clove and ascorbic acid were also added to juices and drinks, and their effect on the content of phenolic compounds in chokeberry juice was investigated. There was similar content of flavonols both in the juices with and without additives—it ranged from about 2.5 to 4 mg/100 mL (Table 2). The relatively low concentration of individual quercetin glycosides (a few milligrams) is typical of chokeberry juices [12,32].

The analysis of the clarification process on the phenolic composition of the juices/drinks was conducted. The clarified and cloudy chokeberry juices/drinks without additives had the following content of anthocyanins before pasteurisation: cyanidin-3-*O*-galactoside—74.88–77.34 mg/100 mL and cyanidin-3-*O*-arabinoside-58.52–60.28 mg/100 mL–the highest concentrations; cyanidin-3-*O*-xyloside-9.19–9.21 mg/100 mL and cyanidin-3-*O*-glucoside-4.16–5.12 mg/100 mL–the lowest concentrations (Table 3). The content of individual cyanidins in chokeberry juices may vary significantly. The ratio of individual cyanidins in juice is not constant, either [33,38,39]. Based on the latest research, the composition of anthocyanins and other polyphenols in chokeberry fruit and products is much more diverse [40,41].

The clarification of the juices by filtration did not cause significant losses of polyphenols. Similarly, White et al. [42] did not observe losses of anthocyanins, flavonols and proanthocyanidins in cranberry juice clarified by sedimentation. At the next stage, extracts of selected plants were added to the juices/drinks. The analysis showed that the addition of spice plant extracts to the unpasteurised juices caused a slight decrease in the anthocyanin content. These changes were minimal. They were statistically significant only when cinnamon extract was added. The addition of the extracts to the cloudy juices/drinks caused a decrease in the content of phenolic acids.

### 3.2. The Effect of the Pasteurisation Process on the Content of Polyphenolic Compounds in Juices/Drinks

The influence of pasteurisation on the content of polyphenolic compounds in the juices/drinks was also analysed. The process (80 °C, 10 min) usually did not change the composition of polyphenols. There were few differences in the concentration of cyanidins, flavanols and phenolic acids as well as frequent changes in the concentration of flavonols (Table 1, Table 2 and Table 3). The pasteurisation of the clarified juices without the additives caused the content of quercetin-3-*O*-vicianoside, galactoside and rutinoside to increase. The pasteurisation of the cloudy juices without the additives reduced the concentration of quercetin-3-*O*-vicianoside. The pasteurised clarified juice with ascorbic acid contained less quercetin-3-*O*-galactoside, rutinoside and glucoside than the corresponding juice/drink before pasteurisation. Wilkes et al. [43] observed that during the pasteurisation of chokeberry juice the content of cyanidins and neochlorogenic acid decreased. There was also a small but statistically significant loss of flavonols. Simultaneously, as the concentration of quercetin glycosides decreased, the concentration of quercetin in the form of aglycon increased.

The thermal treatment (85 °C, 30 min) of pomegranate juice during aseptic filling did not cause a decrease in the content of some anthocyanins, including cyanidin-3-*O*-glucoside, which can also be found in chokeberry juice. However, the total anthocyanin content decreased by 3.92% during heating. The changes were statistically significant. The heating of juice at 95 °C for 45 s caused a decrease in the concentration of all anthocyanins. The total anthocyanin concentration decreased from 269.88 to 230.03 mg/L [44]. Based on the publications cited above, the degradation of anthocyanins during pasteurisation is influenced by raw material, temperature and heating time. The anthocyanin content is sometimes higher in the differential method from HPLC when the results are expressed as equivalents of the same compound. If anthocyanin standards are used in HPLC, the sum of anthocyanins is rather higher than the result obtained by the differential method [45]. In six out of seven juices/drinks using the differential method performed in cuvettes or plates, the content of anthocyanins was lower than the results of measurements from two HPLC systems [46]. Another factor that may affect the anthocyanin transformation during pasteurisation is the method of juice production, including the preparation of raw material, e.g., whole fruit, crushed fruit [41,42]. The influence of the additives (cinnamon and clove extracts) on the polyphenol content was only noticeable in the clarified juices/drinks. It affected only flavonols. The pasteurisation of the clarified juice/drink with the clove extract additive caused a decrease in the content of quercetin-3-*O*-vicianoside and quercetin-3-*O*-galactoside. It also caused an increase in the content of quercetin-3-*O*-glucoside (Table 2). The addition of cinnamon extract during the pasteurisation of the clarified juice/drink caused an increase in the content of quercetin-3-*O*-glycosides (vicianoside, galactoside, glucoside). The pasteurisation (85 °C, 6 min) of chokeberry drinks containing cistus, green tea, and nettle extracts significantly reduced the total anthocyanin content. On the other hand, the total polyphenol content in sucrose-sweetened drinks increased, but it did not change in stevia-sweetened drinks [47].

Talcott and co-workers [48] pasteurised (95 °C, 15 min) muscadine juice and observed changes in its colour and composition as a result of the co-pigmentation of anthocyanins with the components of an aqueous rosemary extract and ascorbic acid. The rosemary extract additive facilitated the formation of co-pigment complexes with anthocyanins and increased the antioxidative activity. The opposite effect was observed in samples with ascorbic acid and rosemary extract, where the content of anthocyanins, ascorbic acid and antioxidative activity decreased. When ascorbic acid was added to the juices/drinks in our experiment, it reduced the content of catechin and chlorogenic acid in the clarified juices/drinks and the content of cyanidin-3-*O*-galactoside and cyanidin-3-*O*-glucoside in the cloudy juices/drinks during pasteurisation.

### 3.3. The Effect of Storage on the Content of Polyphenolic Compounds in Pasteurised Juices/Drinks

The storage of chokeberry juices caused changes in the composition of polyphenols (Table 4, Table 5 and Table 6). The degradation of cyanidins was particularly noticeable, as there were undesirable changes in these compounds every month. Changes in the content of polyphenols, especially the degradation of anthocyanins, are typical of stored juices. They depend on the storage time and temperature [49,50]. Changes in the content of individual groups of polyphenols in the analysed juices/drinks during storage are presented on Figure 1.

To illustrate the influence of the additive on the degree of degradation of anthocyanins, the reaction rate constant *k* for the first-order reaction was determined and the half-life *t*_1/2_ was calculated. The half-life showed that ascorbic acid and the spice plant extracts accelerated the degradation of anthocyanins in the juices (Table 7). Fresh chokeberry juice has high polyphenol content, which is related to potential health-promoting effect; however, processing and storage could often influence polyphenols losses. Ascorbic acid caused the greatest degradation of cyanidins. Among the juices with the extracts, only the clove extract extended the half-life of cyanidins in the cloudy juices. The most stable cyanidins were cyanidin-3-*O*-galactoside, followed by cyanidin-3-*O*-arabinoside, glucoside, and xyloside. The degradation of individual cyanidins occurred more rapidly in the clarified juices, except for the juices with ascorbic acid. The degradation of cyanidin-3-*O*-arabinoside occurred faster in the juices without additives. After three months of storage the anthocyanins loss level ranged from 66.6% for cyanidin-3-*O*-galactoside in the cloudy juice with the clove extract to 82.77% for cyanidin-3-*O*-glucoside in the cloudy juice with ascorbic acid.

After slightly more than 3 months of storage of blueberry and chokeberry nectar, there were considerable losses of anthocyanins observed, i.e., cyanidin-3-*O*-galactoside–67.3%, cyanidin-3-*O*-glucoside–66.1%, and cyanidin-3-*O*-arabinoside–74.6% [51]. The research by Wilkes with colleagues [43], who studied the influence of storage time on the losses of cyanidins in chokeberry juice, showed that the changes were less intense. After three months of storage, the content of cyanidin-3-*O*-galactoside, glucoside and arabinoside in the juice decreased by over 30%, whereas the content of cyanidin-3-*O*-xyloside decreased by about 26%.

Substances made from plants do not always prevent the loss of polyphenols. Stevia, with which chokeberry juices were sweetened, was an additional source of polyphenols, including flavonoids and phenolic acids. During 60-day storage, the losses of total polyphenols, flavonoids and cyanidin-3-*O*-glycosides in juices with stevia were greater than the losses in sucrose-sweetened juice [52]. Extracts of various origins had different effect on the retention of anthocyanins in blackberry juice. 0.1% extracts added to the juice increased the total anthocyanin content by 2%. During the storage of juices for 52 days there was a loss of anthocyanins observed. In comparison with juice without an additive (119.85 mg/L), juices with an olive leaf and pine bark extract PE 5:1 or bioflavonoids contained less anthocyanins (103.44–114.21 mg/L). The content of anthocyanins in juices with the red wine extract PE 4:1 was at a comparable level (118.84 mg/L), but it was greater in the extracts with green tea, pine bark PE 95% and red wine PE 30% (131.99–135.57 mg/L) [53]. The storage of wine causes losses of monomeric anthocyanins and the formation of polymeric pigments mainly between anthocyanins and proanthocyanins. The addition of *Origanum vulgare* and *Satureja thymbra* extracts or rosmarinic acid to wine resulted in the formation of co-pigments, but it did not affect the rate of polymerisation [54].

Plant extracts were added to cherry juice concentrate to improve and increase its colour stability during storage. The addition of pomegranate rind extract or green tea extract increased the stability of the main anthocyanins: cyanidin-3-*O*-glucosylrutinoside, cyanidin-3-*O*-rutinoside and cyanidin-3-*O*-sophoroside. The cherry stem extract reduced the stability of anthocyanins in the concentrate during storage for 110 days [55].

When ascorbic acid was added to the chokeberry juices, it accelerated degradation. The half-life values were 34.5–41.5% lower than in the juices without the additive. The rate of degradation of anthocyanins depends on the concentration of ascorbic acid, the pH of the environment and temperature. The concentration of anthocyanins in sweet potato decreased as the concentration of ascorbic acid increased, and it was clearly noticeable at 25 °C [56]. Farr and Giusti [57] noted high values of *t*_1/2_ = 858 h for the anthocyanin chokeberry extract. When ascorbic acid was added at amounts of 250, 500, and 1000 mg/L, it accelerated the degradation of anthocyanins in the extract, which resulted in lower half-life values, i.e., *t*_1/2_ = 68, 43 and 24 h, respectively. When pH was reduced, it stabilised anthocyanins in the presence of ascorbic acid [58]. The pH of the juices/drinks tested in the study (pH = 3.42–3.52) favoured the stability of anthocyanins and polyphenols (unpublished data). According to Farr and Giusti [57], the reactivity of ascorbic acid with C-4 carbon in anthocyanins is the mechanism catalysing their degradation. When cyanidin-3-*O*-galactoside was synthesised with pyruvic acid, the resulting product was pyranoanthocyanin with blocked C-4 carbon. The half-lives of the tested compounds combined with ascorbic acid at various concentrations were as follows: 5-carboxypyranocyanidin-3-galactoside > chokeberry extract > cyanidin-3-*O*-galactoside. This suggests that C-4 carbon plays an important role in the degradation of anthocyanins under the influence of ascorbic acid. However, the example of the chokeberry extract, whose main anthocyanin is cyanidin-3-*O*-galactoside, shows that this is not the only mechanism. It is very likely that the other polyphenols contained in the extract also limited the degradation of anthocyanins.

The protective effect of other polyphenols could be explained by the research conducted by Stebbins et al. [59]. The authors observed that cyanidin-3-*O*-β-glucoside and malvidin-3-*O*-β-glucoside were oxidised into 6-hydroxy forms in the presence of ascorbic acid. They suggested that the formation of 6-hydroxy-cyanidin3-*O*-β-glucoside and 6-hydroxy-malvidin-3-*O*-β-glucoside occurred under the influence of hydroxyl radicals (HO^•^) generated by the Haber-Weiss reaction, in which ascorbic acid is involved. Electron spin resonance (ESR) analysis in systems with blackberry extract and cyanidin-3-*O*-β-glucoside showed the protective role of other polyphenols contained in the extract, which scavenged hydroxyl radicals (HO^•^).

The protective effect of polyphenols may come from other sources. Roidoung, Dolan and Siddiq [60] added gallic acid to cranberry juice enriched with ascorbic acid and observed lower loss of anthocyanins. On the other hand, it is necessary to select the right raw material and antioxidants to achieve the desired effect. Otherwise, inadequate combinations may provide little or no protection to anthocyanins. Gérard and co-workers [61] studied antioxidants in the form of seven pure compounds and two extracts with strong antioxidative effect, i.e., green coffee bean extract and rosemary extract. The anthocyanin extracts of black carrot, grape juice and purple sweet potato to which 200 mg/L ascorbic acid was added were very slightly stabilised only by chlorogenic acid, sinapic acid, fumaric acid, and β-carotene. There is a study indicating the benefits of using ascorbic acid as an additive. The storage of sour cherry concentrate with ascorbic acid limited the degradation of anthocyanins at temperatures of 4 °C and 24 °C, but at 45 °C it sharply reduced their concentration [62].

## 4. Conclusions

Fresh chokeberry juice has high polyphenol content, which is related to its potential health-promoting effect; however, processing and storage can often influence polyphenol losses. Our study was an attempt to reduce the loss of polyphenol in chokeberry juice by adding aqueous cinnamon and clove extracts. The clarification of the juices did not cause significant changes in the concentration of polyphenols. The addition of the extracts decreased the content of phenolic acids in the cloudy juices. The pasteurisation of the juices usually did not cause changes in the composition of polyphenols, but the cinnamon and clove extracts caused significant differences in the content of flavonols. The degradation of anthocyanins was the main change in the composition of the chokeberry juices observed during storage. Half-life (*t*_1/2_) was used to determine the potential protective influence of the extracts on cyanidin. It showed that the spice plant extracts used as additives accelerated the degradation of anthocyanins in the juices, except the clove extract added to the cloudy juices, which extended the half-life of the cyanidins analysed in the study. The research results showed that despite the common view about the beneficial effects of polyphenols and other compounds exhibiting antioxidative potential to each other, it is very difficult to achieve protection from degradation. Further research is necessary to investigate this problem.

## Figures and Tables

**Figure 1 antioxidants-09-00801-f001:**
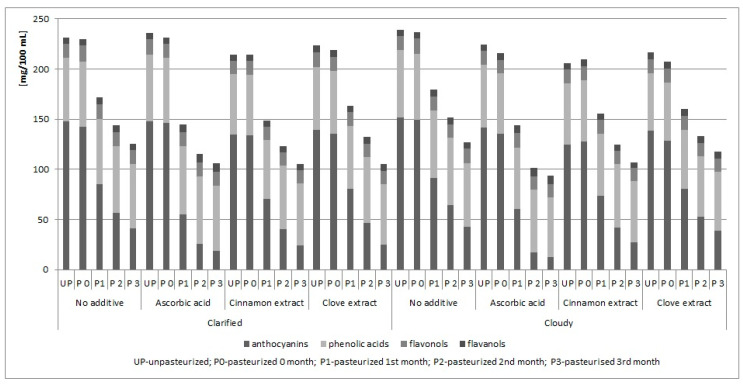
Changes in the content of individual groups of polyphenols in the analysed juices/drinks during storage.

**Table 1 antioxidants-09-00801-t001:** Phenolic acids and flavanols content before and after juice/drink pasteurisation.

	Juice/Drink		
	Additive	Unpasteurised	Pasteurised
(+)-catechin [mg/100 mL]			
Clarified	No additives	1.57 ± 0.00 ^A,a^^,^^b^	1.51 ± 0.09 ^A,b,c^
	Ascorbic acid	1.70 ± 0.02 ^A,a,b^	1.60 ± 0.03 ^B,a,b^
	Cinnamon extract	1.57 ± 0.15 ^A,a,b^	1.39 ± 0.01 ^A,c^
	Clove extract	1.81 ± 0.02 ^A,a^	1.73 ± 0.04 ^A,a^
Cloudy	No additives	1.58 ± 0.02 ^A,a,b^	1.52 ± 0.06 ^A,b,c^
	Ascorbic acid	1.66 ± 0.01 ^A,a,b^	1.75 ± 0.09 ^A,a^
	Cinnamon extract	1.45 ± 0.04 ^A,b^	1.48 ± 0.02 ^A,b,c^
	Clove extract	1.70 ± 0.12 ^A,a,b^	1.77 ± 0.11 ^A,a^
(−)-epicatechin [mg/100 mL]			
Clarified	No additives	4.42 ± 0.02 ^A,b^	4.53 ± 0.07 ^A,c^
	Ascorbic acid	4.50 ± 0.08 ^A,b^	4.53 ± 0.06 ^A,c^
	Cinnamon extract	4.63 ± 0.11 ^A,a,b^	4.82 ± 0.04 ^A,a^
	Clove extract	4.74 ± 0.07 ^A,a,b^	4.79 ± 0.03 ^A,a,b^
Cloudy	No additives	4.65 ± 0.02 ^A,a,b^	4.66 ± 0.14 ^A,a,b,c^
	Ascorbic acid	4.44 ± 0.02 ^A,b^	4.60 ± 0.09 ^A,b,c^
	Cinnamon extract	4.62 ± 0.04 ^A,a,b^	4.82 ± 0.05 ^A,a^
	Clove extract	4.94 ± 0.18 ^A,a^	4.86 ± 0.02 ^A,a^
neochlorogenic acid [mg/100 mL]			
Clarified	No additives	33.21 ± 0.51 ^A,a,b^	34.05 ± 0.56 ^A,a^
	Ascorbic acid	34.77 ± 0.89 ^A,a^	33.86 ± 0.33 ^A,a,b^
	Cinnamon extract	31.59 ± 0.33 ^A,b,c^	31.76 ± 0.68 ^A,b,c^
	Clove extract	32.91 ± 0.10 ^A,a,b^	32.42 ± 0.63 ^A,a,b,c^
Cloudy	No additives	34.70 ± 1.21 ^A,a^	34.19 ± 0.64 ^A,a^
	Ascorbic acid	32.80 ± 0.07 ^A,a,b^	31.22 ± 1.19 ^A,c^
	Cinnamon extract	31.82 ± 0.37 ^A,b,c^	31.87 ± 0.46 ^A,b,c^
	Clove extract	30.03 ± 0.72 ^A,c^	30.81 ± 1.06 ^A,c^
chlorogenic acid [mg/100 mL]			
Clarified	No additives	30.30 ± 0.04 ^A,a,b,c^	30.97 ± 0.35 ^A,a^
	Ascorbic acid	32.13 ± 0.51 ^A,a^	30.82 ± 0.34 ^B,a^
	Cinnamon extract	28.94 ± 0.32 ^A,c,d^	28.98 ± 0.27 ^A,b^
	Clove extract	30.23 ± 0.05 ^A,b,c^	30.08 ± 0.46 ^A,a,b^
Cloudy	No additives	31.98 ± 1.14 ^A,a,b^	31.15 ± 0.64 ^A,a^
	Ascorbic acid	29.39 ± 0.11 ^A,c^	28.93 ± 0.35 ^A,b^
	Cinnamon extract	29.22 ± 0.06 ^A,c^	29.25 ± 0.32 ^A,b^
	Clove extract	27.28 ± 0.22 ^A,d^	27.28 ± 0.54 ^A,c^

a, b, c, d—means in a column followed by the same small letter are not significantly different (*p* > 0.05); A, B—means in a row followed by the same capital letter are not significantly different (*p* > 0.05); values are means of three determinations ± SD.

**Table 2 antioxidants-09-00801-t002:** Flavonol content before and after juice/drink pasteurisation.

		Juice/Drink	
	Additive	Unpasteurised	Pasteurised
quercetin-3-*O*-vicianoside [mg/100 mL]			
Clarified	No additives	4.00 ± 0.07 ^B,a^	4.86 ± 0.06 ^A,a^
	Ascorbic acid	4.07 ± 0.10 ^A,a^	3.90 ± 0.05 ^A,b,c^
	Cinnamon extract	3.68 ± 0.02 ^B,b^	3.76 ± 0.02 ^A,b,c^
	Clove extract	3.89 ± 0.00 ^A,a^	3.79 ± 0.02 ^B,b,c^
Cloudy	No additives	3.90 ± 0.07 ^B,a^	4.81 ± 0.12 ^A,a^
	Ascorbic acid	3.86 ± 0.03 ^A,a,b^	3.82 ± 0.11 ^A,b,c^
	Cinnamon extract	3.91 ± 0.02 ^A,a^	3.99 ± 0.09 ^A,b^
	Clove extract	3.65 ± 0.01 ^A,b^	3.68 ± 0.21 ^A,c^
quercetin-3-*O*-galactoside [mg/100 mL]			
Clarified	No additives	3.98 ± 0.05 ^B,a,b^	4.28 ± 0.08 ^A,a^
	Ascorbic acid	4.30 ± 0.02 ^A,a^	3.96 ± 0.06 ^B,b^
	Cinnamon extract	3.60 ± 0.01 ^B,b^	3.71 ± 0.03 ^A,b,c^
	Clove extract	3.87 ± 0.01 ^A,a,b^	3.76 ± 0.01 ^B,b,c^
Cloudy	No additives	4.00 ± 0.22 ^A,a,b^	4.27 ± 0.12 ^A,a^
	Ascorbic acid	3.99 ± 0.21 ^A,a,b^	3.81 ± 0.12 ^A,b,c^
	Cinnamon extract	3.77 ± 0.04 ^A,b^	3.87 ± 0.07 ^A,b,c^
	Clove extract	3.59 ± 0.03 ^A,b^	3.59 ± 0.21 ^A,c^
quercetin-3-*O*-rutinoside [mg/100 mL]			
Clarified	No additives	3.76± 0.09 ^B,a,b^	4.11 ± 0.04 ^A,a^
	Ascorbic acid	4.10± 0.10 ^A,a^	3.71 ± 0.06 ^B,b^
	Cinnamon extract	3.44± 0.04 ^A,b^	3.54 ± 0.04 ^A,c^
	Clove extract	3.93± 0.02 ^A,a,b^	3.89 ± 0.04 ^A,a,b,c^
Cloudy	No additives	3.90± 0.34 ^A,a,b^	4.12 ± 0.10 ^A,a^
	Ascorbic acid	3.90± 0.24 ^A,a,b^	3.71 ± 0.13 ^A,b,c^
	Cinnamon extract	3.73± 0.01 ^A,a,b^	3.77 ± 0.09 ^A,a,b,c^
	Clove extract	3.79± 0.03 ^A,a,b^	4.03 ± 0.31 ^A,a,b^
quercetin 3-*O*-glucoside [mg/100 mL]			
Clarified	No additives	2.51 ± 0.05 ^A,a,b,c^	2.52 ± 0.05 ^A,b,c^
	Ascorbic acid	2.76 ± 0.05 ^A,a,b^	2.53 ± 0.03 ^B,b,c^
	Cinnamon extract	2.28 ± 0.03 ^B,c^	2.37 ± 0.01 ^A,c^
	Clove extract	2.80 ± 0.01 ^A,a^	2.74 ± 0.01 ^B,a^
Cloudy	No additives	2.62 ± 0.16 ^A,a,b,c^	2.54 ± 0.06 ^A,b,c^
	Ascorbic acid	2.61 ± 0.18 ^A,a,b,c^	2.48 ± 0.08 ^A,b,c^
	Cinnamon extract	2.41 ± 0.03 ^A,b,c^	2.47 ± 0.04 ^A,b,c^
	Clove extract	2.62 ± 0.08 ^A,a,b^	2.62 ± 0.14 ^A,a,b^

a, b, c—means in a column followed by the same small letter are not significantly different (*p* > 0.05); A, B—means in a row followed by the same capital letter are not significantly different (*p* > 0.05); values are means of three determinations ± SD.

**Table 3 antioxidants-09-00801-t003:** Cyanidin content before and after juice/drink pasteurisation.

		Juice/Drink	
	Additive	Unpasteurised	Pasteurised
cyanidin-3-*O*-galactoside [mg/100 mL]			
Clarified	No additives	74.88 ± 0.63 ^A,a^	73.93 ± 2.32 ^A,a,b^
	Ascorbic acid	74.87 ± 3.05 ^A,a^	74.33 ± 1.45 ^A,a,b^
	Cinnamon extract	68.65 ± 2.61 ^A,a,b^	68.44 ± 1.12 ^A,b,c^
	Clove extract	70.47 ± 1.41 ^A,a,b^	69.15 ± 1.31 ^A,b,c^
Cloudy	No additives	77.34 ± 3.94 ^A,a^	76.91 ± 1.92 ^A,a^
	Ascorbic acid	72.23 ± 0.86 ^A,a,b^	69.58 ± 0.46 ^B,b,c^
	Cinnamon extract	64.69 ± 0.96 ^A,b^	64.10 ± 2.30 ^A,c^
	Clove extract	70.33 ± 2.12 ^A,a,b^	66.07 ± 4.08 ^A,c^
cyanidin-3-*O*-glucoside [mg/100 mL]			
Clarified	No additives	5.03 ± 0.00 ^A,a^	4.91 ± 0.22 ^A,a,b^
	Ascorbic acid	5.03 ± 0.23 ^A,a^	4.92 ± 0.12 ^A,a,b^
	Cinnamon extract	4.56 ± 0.21 ^A,a,b^	4.48 ± 0.06 ^A,b,c^
	Clove extract	4.69 ± 0.01 ^A,a,b^	4.59 ± 0.10 ^A,b,c^
Cloudy	No additives	5.12 ± 0.29 ^A,a^	5.16 ± 0.16 ^A,a^
	Ascorbic acid	4.79 ± 0.03 ^A,a,b^	4.47 ± 0.01 ^B,b,c^
	Cinnamon extract	4.16 ± 0.09 ^A,b^	4.35 ± 0.11 ^A,c^
	Clove extract	4.64 ± 0.13 ^A,a,b^	4.23 ± 0.33 ^A,c^
cyanidin-3-*O*-arabinoside [mg/100 mL]			
Clarified	No additives	58.52 ± 0.76 ^A,a^	54.40 ± 2.66 ^A,a,b^
	Ascorbic acid	58.40 ± 2.49 ^A,a^	58.04 ± 1.49 ^A,a^
	Cinnamon extract	53.07 ± 2.27 ^A,a,b^	52.54 ± 0.83 ^A,a,b^
	Clove extract	54.97 ± 0.84 ^A,a,b^	53.31 ± 1.42 ^A,a,b^
Cloudy	No additives	60.28 ± 3.70 ^A,a^	57.63 ± 1.97 ^A,a^
	Ascorbic acid	56.06 ± 1.39 ^A,a,b^	52.22 ± 2.16 ^A,a,b^
	Cinnamon extract	48.25 ± 1.57 ^A,b^	51.23 ± 2.76 ^A,b^
	Clove extract	55.03 ± 3.05 ^A,a,b^	49.83 ± 3.46 ^A,b^
cyanidin-3-*O*-xyloside [mg/100 mL]			
Clarified	No additives	9.21 ± 0.21 ^A,a,b^	9.25 ± 0.42 ^A,a,b^
	Ascorbic acid	9.43 ± 0.53 ^A,a^	9.22 ± 0.32 ^A,a,b^
	Cinnamon extract	8.29 ± 0.31 ^A,b,c^	8.29 ± 0.15 ^A,b^
	Clove extract	9.00 ± 0.05 ^A,a,b^	8.59 ± 0.26 ^A,b^
Cloudy	No additives	9.19 ± 0.11 ^A,a,b^	9.82 ± 0.37 ^A,a^
	Ascorbic acid	8.60 ± 0.10 ^A,a,b,c^	8.97 ± 0.27 ^A,a,b^
	Cinnamon extract	7.81 ± 0.25 ^A,c^	8.08 ± 0.25 ^A,b^
	Clove extract	8.65 ± 0.35 ^A,a,b,c^	8.35 ± 0.91 ^A,b^

a, b, c—means in a column followed by the same small letter are not significantly different (*p* > 0.05); A, B—means in a row followed by the same capital letter are not significantly different (*p* > 0.05); values are means of three determinations ± SD.

**Table 4 antioxidants-09-00801-t004:** Changes in the content of phenolic acids and flavanols during storage of tested juices/drinks.

		Juice/Drink			
		Storage [Month]			
	Additive	0	1st	2nd	3rd
(+)-catechin [mg/100 mL]					
Clarified	No additives	1.51 ± 0.09 ^B,b,c^	1.96 ± 0.09 ^A,b,c^	2.01 ± 0.03 ^A,d^	2.04 ± 0.07 ^A,bc^
	Ascorbic acid	1.60 ± 0.03 ^C,a,b^	2.91 ± 0.03 ^B,a^	3.47 ± 0.04 ^A,b^	3.57 ± 0.08 ^A,a^
	Cinnamon extract	1.39 ± 0.01 ^C,c^	1.70 ± 0.06 ^B,d^	1.84 ± 0.02 ^A,e^	1.80 ± 0.04 ^A,B,c^
	Clove extract	1.73 ± 0.04 ^B,a^	1.99 ± 0.03 ^A,b,c^	2.05 ± 0.00 ^A,d^	1.97 ± 0.11 ^A,b,c^
Cloudy	No additives	1.52 ± 0.06 ^C,b,c^	1.87 ± 0.03 ^B,c,d^	2.05 ± 0.02 ^A,d^	2.04 ± 0.06 ^A,b,c^
	Ascorbic acid	1.75 ± 0.09 ^C,a^	2.98 ± 0.08 ^B,a^	3.69 ± 0.03 ^A,a^	3.54 ± 0.26 ^A,a^
	Cinnamon extract	1.48 ± 0.02 ^B,b,c^	1.81 ± 0.10 ^A,c,d^	1.82 ± 0.01 ^A,e^	1.82 ± 0.00 ^A,c^
	Clove extract	1.77 ± 0.11 ^B,a^	2.09 ± 0.07 ^A,b^	2.14 ± 0.04 ^A,c^	2.19 ± 0.13 ^A,b^
(−)-epicatechin [mg/100 mL]					
Clarified	No additives	4.53 ± 0.07 ^A,c^	4.39 ± 0.15 ^A,b,c^	4.52 ± 0.16 ^A,b^	4.41 ± 0.18 ^A,a,b,c^
	Ascorbic acid	4.53 ± 0.06 ^B,c^	4.75 ± 0.06 ^A,a^	4.76 ± 0.06 ^A,a^	4.83 ± 0.08 ^A,a^
	Cinnamon extract	4.82 ± 0.04 ^A,a^	4.31 ± 0.08 ^B,c^	4.26 ± 0.02 ^B,c^	4.05 ± 0.09 ^C,c^
	Clove extract	4.79 ± 0.03 ^A,a,b^	4.66 ± 0.03 ^A,B,a,b^	4.52 ± 0.00 ^B,b^	4.19 ± 0.19 ^C,b,c^
Cloudy	No additives	4.66 ± 0.14 ^A,a,b,c^	4.53 ± 0.10 ^A,a,b,c^	4.61 ± 0.03 ^A,a,b^	4.35 ± 0.31 ^A,a,b,c^
	Ascorbic acid	4.60 ± 0.09 ^A,b,c^	4.68 ± 0.17 ^A,a,b^	4.73 ± 0.06 ^A,a^	4.58 ± 0.23 ^A,a,b^
	Cinnamon extract	4.82 ± 0.05 ^A,a^	4.45 ± 0.06 ^B,a,b,c^	4.28 ± 0.01 ^C,c^	4.10 ± 0.04 ^D,b,c^
	Clove extract	4.86 ± 0.02 ^A,a^	4.55 ± 0.15 ^A,a,b,c^	4.61 ± 0.02 ^A,a,b^	4.54 ± 0.19 ^A,a,b,c^
neochlorogenic acid [mg/100 mL]					
Clarified	No additives	34.05 ± 0.56 ^A,a^	34.02 ± 0.99 ^A,a^	36.12 ± 0.39 ^A,a^	34.79 ± 1.30 ^A,a^
	Ascorbic acid	33.86 ± 0.33 ^C,a,b^	36.89 ± 2.46 ^A,a^	36.26 ± 0.30 ^A,B,a^	34.92 ± 1.00 ^B,C,a^
	Cinnamon extract	31.76 ± 0.68 ^A,B,b,c^	31.30 ± 2.89 ^B,a^	34.50 ± 0.84 ^A,b,c^	33.83 ± 0.91 ^A,B,a^
	Clove extract	32.42 ± 0.63 ^A,a,b,c^	33.04 ± 2.33 ^A,a^	35.66 ± 0.41 ^A,a,b^	32.58 ± 1.31 ^A,a^
Cloudy	No additives	34.19 ± 0.64 ^A,a^	36.85 ± 2.88 ^A,a^	36.72 ± 0.34 ^A,a^	34.21 ± 2.27 ^A,a^
	Ascorbic acid	31.22 ± 1.19 ^A,c^	32.71 ± 0.09 ^A,a^	33.85 ± 0.44 ^A,c,d^	32.77 ± 2.03 ^A,a^
	Cinnamon extract	31.87 ± 0.46 ^A,b,c^	32.30 ± 1.98 ^A,a^	34.63 ± 0.25 ^A,b,c^	33.56 ± 0.95 ^A,a^
	Clove extract	30.81 ± 1.06 ^A,c^	31.40 ± 3.27 ^A,a^	32.95 ± 0.43 ^A,d^	32.63 ± 0.46 ^A,a^
chlorogenic acid [mg/100 mL]					
Clarified	No additives	30.97 ± 0.35 ^A,a^	30.82 ± 0.33 ^A,B,a,b^	30.32 ± 0.12 ^A,B,a,b^	29.28 ± 1.09 ^A,B,a,b^
	Ascorbic acid	30.82 ± 0.34 ^A,B,a^	31.20 ± 0.02 ^A,a^	30.74 ± 0.23 ^A,B,a^	29.94 ± 0.73 ^B,a^
	Cinnamon extract	28.98 ± 0.27 ^A,b^	27.78 ± 0.44 ^B,d,e^	29.04 ± 0.26 ^A,c^	28.46 ± 0.52 ^A,B,a,b^
	Clove extract	30.08 ± 0.46 ^A,a,b^	29.51 ± 1.12 ^A,a,b,c^	29.98 ± 0.16 ^A,b^	27.96 ± 1.24 ^A,a,b^
Cloudy	No additives	31.15 ± 0.64 ^A,a^	30.79 ± 0.61 ^A,B,a,b^	30.77 ± 0.08 ^A,B,a^	29.38 ± 0.89 ^B,a,b^
	Ascorbic acid	28.93 ± 0.35 ^A,b^	28.95 ± 0.54 ^A,c,d^	28.38 ± 0.21 ^A,d^	27.21 ± 1.73 ^A,a,b^
	Cinnamon extract	29.25 ± 0.32 ^A,b^	29.48 ± 0.34 ^A,b,c^	28.84 ± 0.08 ^A,B,c,d^	27.98 ± 0.79 ^B,a,b^
	Clove extract	27.28 ± 0.54 ^A,c^	27.17 ± 0.71 ^A,e^	27.14 ± 0.19 ^A,e^	26.82 ± 0.44 ^A,b^

a, b, c, d, e—means in a column followed by the same small letter are not significantly different (*p* > 0.05); A, B, C, D—means in a row followed by the same capital letter are not significantly different (*p* > 0.05); values are means of three determinations ± SD.

**Table 5 antioxidants-09-00801-t005:** Changes in flavonol content during storage of tested juices/drinks.

		Juice/Drink			
		Storage [month]			
	Additive	0	1st	2nd	3rd
quercetin-3-*O*-vicianoside [mg/100 mL]					
Clarified	No additives	4.86 ± 0.06 ^A,a^	4.65 ± 0.10 ^B,a^	4.36 ± 0.08 ^C,a^	4.25 ± 0.09 ^C,a^
	Ascorbic acid	3.90 ± 0.05 ^B,b,c^	4.38 ± 0.07 ^A,a,b^	4.36 ± 0.06 ^A,a^	4.32 ± 0.13 ^A,a^
	Cinnamon extract	3.76 ± 0.02 ^B,b,c^	3.74 ± 0.23 ^B,c^	4.13 ± 0.06 ^A,a,b^	3.98 ± 0.14 ^A,B,a^
	Clove extract	3.79 ± 0.02 ^B^^,^^C,b,c^	3.68 ± 0.03 ^C,c^	4.17 ± 0.06 ^A,a,b^	3.95 ± 0.15 ^A,B,a^
Cloudy	No additives	4.81 ± 0.12 ^A,a^	4.41 ± 0.07 ^A,B,a,b^	4.23 ± 0.34 ^B,a,b^	4.39 ± 0.23 ^A,B,a^
	Ascorbic acid	3.82 ± 0.11 ^B,b,c^	4.54 ± 0.19 ^A,a,b^	4.24 ± 0.01 ^AB,a,b^	4.06 ± 0.28 ^B,a^
	Cinnamon extract	3.99 ± 0.09 ^B,b^	4.38 ± 0.09 ^A,a,b^	4.22 ± 0.01 ^A,B,a,b^	4.01 ± 0.25 ^A,B,a^
	Clove extract	3.68 ± 0.21 ^B,c^	4.22 ± 0.11 ^A,b^	3.99 ± 0.00 ^A,B,b^	3.94 ± 0.09 ^A,B,a^
quercetin-3-*O*-galactoside [mg/100 mL]					
Clarified	No additives	4.28 ± 0.08 ^A,a^	4.05 ± 0.06 ^A,B,a^	3.76 ± 0.12 ^C,a,b^	3.77 ± 0.15 ^B,C,a^
	Ascorbic acid	3.96 ± 0.06 ^A,b^	4.05 ± 0.11 ^A,a^	3.92 ± 0.05 ^A,a^	3.83 ± 0.16 ^A,a^
	Cinnamon extract	3.71 ± 0.03 ^A,b,c^	3.65 ± 0.15 ^A,B,c,d^	3.54 ± 0.05 ^A,B,c,d^	3.44 ± 0.10 ^B,a^
	Clove extract	3.76 ± 0.01 ^A,b,c^	3.69 ± 0.07 ^A,c,d^	3.62 ± 0.09 ^A,b,c^	3.64 ± 0.28 ^A,a^
Cloudy	No additives	4.27 ± 0.12 ^A,a^	3.86 ± 0.05 ^B,a,b,c^	3.85 ± 0.01 ^B,a^	3.78 ± 0.17 ^B,a^
	Ascorbic acid	3.81 ± 0.12 ^A,b,c^	3.97 ± 0.11 ^A,a,b^	3.76 ± 0.02 ^A,a,b^	3.57 ± 0.28 ^A,a^
	Cinnamon extract	3.87 ± 0.07 ^A,b,c^	3.73 ± 0.09 ^A,B,b,c,d^	3.59 ± 0.02 ^B,C,b,c,d^	3.45 ± 0.08 ^C,a^
	Clove extract	3.59 ± 0.21 ^A,c^	3.51 ± 0.07 ^A,d^	3.45 ± 0.01 ^A,d^	3.39 ± 0.04 ^A,a^
quercetin-3-*O*-rutinoside [mg/100 mL]					
Clarified	No additives	4.11 ± 0.04 ^A,a^	3.82 ± 0.29 ^A,a,b^	3.37 ± 0.08 ^B,a^	3.31 ± 0.16 ^B,a^
	Ascorbic acid	3.71 ± 0.06 ^A,bc^	3.43 ± 0.07 ^B,a,b^	3.40 ± 0.03 ^B,a^	3.47 ± 0.06 ^B,a^
	Cinnamon extract	3.54 ± 0.04 ^A,c^	3.27 ± 0.19 ^A,B,b^	3.23 ± 0.03 ^B,a^	3.18 ± 0.08 ^B,a^
	Clove extract	3.89 ± 0.04 ^A,a,b,c^	3.61 ± 0.27 ^A,B,a,b^	3.26 ± 0.02 ^B,C,a^	3.10 ± 0.07 ^C,a^
Cloudy	No additives	4.12 ± 0.10 ^A,a^	3.43 ± 0.02 ^B,a,b^	3.25 ± 0.34 ^B,a^	3.51 ± 0.37 ^A,B,a^
	Ascorbic acid	3.71 ± 0.13 ^A,b,c^	3.70 ± 0.27 ^A,a,b^	3.37 ± 0.00 ^A,a^	3.27 ± 0.31 ^A,a^
	Cinnamon extract	3.77 ± 0.09 ^A,a,b,c^	3.52 ± 0.20 ^A,B,a,b^	3.24 ± 0.02 ^B,a^	3.11 ± 0.29 ^B,a^
	Clove extract	4.03 ± 0.31 ^A,a,b^	3.89 ± 0.10 ^A,a^	3.19 ± 0.01 ^B,a^	3.21 ± 0.03 ^B,a^
quercetin-3-*O*-glucoside [mg/100 mL]					
Clarified	No additives	2.52 ± 0.05 ^A,b,c^	2.49 ± 0.02 ^A,a,b^	2.39 ± 0.04 ^A,B,b,c^	2.31 ± 0.10 ^B,b,c,d^
	Ascorbic acid	2.53 ± 0.03 ^A,b,c^	2.50 ± 0.02 ^A,a,b^	2.47 ± 0.02 ^A,a,b,c^	2.49 ± 0.03 ^A,a,b,c^
	Cinnamon extract	2.37 ± 0.01 ^A,c^	2.21 ± 0.13 ^A,c^	2.24 ± 0.02 ^A,c^	2.21 ± 0.04 ^A,c,d^
	Clove extract	2.74 ± 0.01 ^A,a^	2.72 ± 0.04 ^A,a^	2.74 ± 0.01 ^A,a^	2.55 ± 0.08 ^B,a,b^
Cloudy	No additives	2.54 ± 0.06 ^A,b,c^	2.40 ± 0.03 ^A,b,c^	2.26 ± 0.27 ^A,c^	2.35 ± 0.06 ^A,a,b,c,d^
	Ascorbic acid	2.48 ± 0.08 ^A,b,c^	2.37 ± 0.15 ^A,b,c^	2.42 ± 0.00 ^A,b,c^	2.33 ± 0.16 ^A,a,b,c,d^
	Cinnamon extract	2.47 ± 0.04 ^A,b,c^	2.24 ± 0.04 ^A,B,c^	2.25 ± 0.01 ^A,B,c^	2.10 ± 0.21 ^B,d^
	Clove extract	2.62 ± 0.14 ^A,a,b^	2.59 ± 0.09 ^A,a,b^	2.64 ± 0.01 ^A,a,b^	2.63 ± 0.03 ^A,a^

a, b, c, d—means in a column followed by the same small letter are not significantly different (*p* > 0.05); A, B, C—means in a row followed by the same capital letter are not significantly different (*p* > 0.05); values are means of three determinations ± SD.

**Table 6 antioxidants-09-00801-t006:** Changes in anthocyanin content during storage of tested juices/drinks.

		Juice/Drink			
		Storage [Month]			
	Additive	0	1st	2nd	3rd
cyaniding-3-*O*-galactoside [mg/100 mL]					
Clarified	No additives	73.93 ± 2.32 ^A,a,b^	46.98 ± 0.95 ^B,a,b^	32.43 ± 4.45 ^C,a,b^	23.83 ± 2.25 ^D,a,b^
	Ascorbic acid	74.33 ± 1.45 ^A,a,b^	30.68 ± 0.51 ^B,e^	14.57 ± 2.81 ^C,d^	10.96 ± 3.09 ^C,c^
	Cinnamon extract	68.44 ± 1.12 ^A,b,c^	39.06 ± 2.66 ^B,d^	23.44 ± 0.63 ^C,c^	14.02 ± 0.84 ^D,b,c^
	Clove extract	69.15 ± 1.31 ^A,b,c^	43.92 ± 1.59 ^B,b,c^	26.20 ± 0.33 ^C,b,c^	14.27 ± 2.05 ^D,b,c^
Cloudy	No additives	76.91 ± 1.92 ^A,a^	50.33 ± 0.81 ^B,a^	36.30 ± 1.34 ^C,a^	25.01 ± 5.83 ^D,a^
	Ascorbic acid	69.58 ± 0.46 ^A,b,c^	33.76 ± 2.03 ^B,e^	9.48 ± 1.84 ^C,d^	6.87 ± 1.90 ^C,c^
	Cinnamon extract	64.10 ± 2.30 ^A,c^	41.40 ± 0.63 ^B,c,d^	24.17 ± 0.50 ^C,c^	15.77 ± 0.82 ^D,a,b,c^
	Clove extract	66.07 ± 4.08 ^A,c^	43.86 ± 2.65 ^B,b,c^	30.07 ± 4.08 ^C,a,b,c^	22.10 ± 7.32 ^C,a,b^
cyanidin-3-*O*-glucoside [mg/100 mL]					
Clarified	No additives	4.91 ± 0.22 ^A,a,b^	2.59 ± 0.11 ^B,a^	1.62 ± 0.26 ^C,a,b^	1.12 ± 0.07 ^D,a,b^
	Ascorbic acid	4.92 ± 0.12 ^A,a,b^	1.30 ± 0.03 ^B,e^	0.60 ± 0.12 ^C,d^	0.44 ± 0.14 ^C,c^
	Cinnamon extract	4.48 ± 0.06 ^A,b,c^	2.16 ± 0.03 ^B,c^	1.14 ± 0.03 ^C,c^	0.67 ± 0.04 ^D,b,c^
	Clove extract	4.59 ± 0.10 ^A,b,c^	2.50 ± 0.10 ^B,a,b^	1.29 ± 0.01 ^C,b,c^	0.70 ± 0.09 ^D,b,c^
Cloudy	No additives	5.16 ± 0.16 ^A,a^	2.61 ± 0.05 ^B,a^	1.79 ± 0.10 ^C,a^	1.22 ± 0.28 ^D,a^
	Ascorbic acid	4.47 ± 0.01 ^A,b,c^	1.59 ± 0.15 ^B,d^	0.38 ± 0.07 ^C,d^	0.28 ± 0.07 ^C,c^
	Cinnamon extract	4.35 ± 0.11 ^A,c^	2.24 ± 0.09 ^B,b,c^	1.18 ± 0.03 ^C,c^	0.77 ± 0.03 ^D,a,b,c^
	Clove extract	4.23 ± 0.33 ^A,c^	2.44 ± 0.12 ^B,a,b^	1.47 ± 0.22 ^C,a,b,c^	1.07 ± 0.36 ^D,a,b^
cyanidin-3-*O*-arabinoside [mg/100 mL]					
Clarified	No additives	54.40 ± 2.66 ^A,a,b^	30.82 ± 1.23 ^B,a,b^	19.97 ± 3.22 ^C,a,b^	14.17 ± 1.68 ^C,a^
	Ascorbic acid	58.04 ± 1.49 ^A,a^	20.01 ± 0.17 ^B,d^	9.43 ± 1.74 ^C,e,f^	6.64 ± 1.99 ^C,b,c^
	Cinnamon extract	52.54 ± 0.83 ^A,a,b^	25.14 ± 2.01 ^B,c^	13.77 ± 0.48 ^C,d,e^	8.05 ± 0.35 ^D,a,b,c^
	Clove extract	53.31 ± 1.42 ^A,a,b^	29.75 ± 0.53 ^B,b^	16.39 ± 0.20 ^C,b,c,d^	8.66 ± 1.23 ^D,a,b,c^
Cloudy	No additives	57.63 ± 1.97 ^A,a^	33.29 ± 0.70 ^B,a^	22.65 ± 0.86 ^C,a^	14.38 ± 4.15 ^D,a^
	Ascorbic acid	52.22 ± 2.16 ^A,a,b^	21.58 ± 1.23 ^B,d^	6.41 ± 1.04 ^C,f^	4.48 ± 0.94 ^C,c^
	Cinnamon extract	51.23 ± 2.76 ^A,b^	26.15 ± 0.87 ^B,c^	14.23 ± 0.39 ^C,c,d,e^	9.00 ± 0.39 ^D,a,b,c^
	Clove extract	49.83 ± 3.46 ^A,b^	29.63 ± 1.32 ^B,b^	18.72 ± 2.80 ^C,a,b,c^	13.30 ± 4.27 ^C,a,b^
cyanidin-3-*O*-xyloside [mg/100 mL]					
Clarified	No additives	9.25 ± 0.42 ^A,a,b^	4.75 ± 0.26 ^B,a^	2.81 ± 0.53 ^C,a,b^	1.93 ± 0.19 ^C,a,b^
	Ascorbic acid	9.22 ± 0.32 ^A,a,b^	2.68 ± 0.04 ^B,e^	1.20 ± 0.22 ^C,d,e^	0.84 ± 0.28 ^C,c^
	Cinnamon extract	8.29 ± 0.15 ^A,b^	3.74 ± 0.30 ^B,c,d^	1.91 ± 0.08 ^C,c,d^	1.08 ± 0.06 ^D,b,c^
	Clove extract	8.59 ± 0.26 ^A,b^	4.45 ± 0.10 ^B,a,b^	2.30 ± 0.05 ^C,b,c^	1.21 ± 0.15 ^D,a,b,c^
Cloudy	No additives	9.82 ± 0.37 ^A,a^	4.79 ± 0.09 ^B,a^	3.17 ± 0.18 ^C,a^	2.08 ± 0.46 ^D,a^
	Ascorbic acid	8.97 ± 0.27 ^A,a,b^	3.18 ± 0.27 ^B,d,e^	0.82 ± 0.13 ^C,e^	0.57 ± 0.10 ^C,c^
	Cinnamon extract	8.08 ± 0.25 ^A,b^	4.06 ± 0.20 ^B,b,c^	1.97 ± 0.05 ^C,c^	1.22 ± 0.06 ^D,a,b,c^
	Clove extract	8.35 ± 0.91 ^A,b^	4.66 ± 0.32 ^B,a,b^	2.66 ± 0.42 ^C,a,b,c^	1.86 ±0.64 ^C,a,b^

a, b, c, d, e, f—means in a column followed by the same small letter are not significantly different (*p* > 0.05); A, B, C, D—means in a row followed by the same capital letter are not significantly different (*p* > 0.05); values are means of three determinations ± SD.

**Table 7 antioxidants-09-00801-t007:** Equation of the curve for degradation, constant reaction rate and half-life of anthocyanins in chokeberry juices.

	Juice/Drink	Equation for the Curve of the Logarithm of Substrate Concentration versus Time *t* [d] for First-Order Reactions	R^2^	*k* [d^-1^]	t_1/2_ [d]
cyanidin-3-*O*-galactoside					
Clarified	No additives	y= −0.0055x + 1.8526	0.9926	5.5 × 10^−3^	126.0
	Ascorbic acid	y= −0.0094x + 1.8129	0.9567	9.4 × 10^−3^	73.7
	Cinnamon extract	y= −0.0076x + 1.8290	0.9995	7.6 × 10^−3^	91.2
	Clove extract	y= −0.0076x + 1.8558	0.9957	7.6 × 10^−3^	91.2
Cloudy	No additives	y= −0.0054x + 1.8773	0.9975	5.4 × 10^−3^	128.3
	Ascorbic acid	y= −0.0119x + 1.8314	0.9570	11.9 × 10^−3^	58.2
	Cinnamon extract	y= −0.0069x + 1.8103	0.9980	6.9 × 10^−3^	100.4
	Clove extract	y= −0.0053x + 1.8097	0.9960	5.3 × 10^−3^	130.8
cyanidin−3-*O*-glucoside					
Clarified	No additives	y= −0.0071x + 0.6602	0.9852	7.1 × 10^−3^	97.6
	Ascorbic acid	y= −0.0116x + 0.5786	0.9244	11.6 × 10^−3^	59.7
	Cinnamon extract	y= −0.0092x + 0.6300	0.9950	9.2 × 10^−3^	75.3
	Clove extract	y= −0.0091x + 0.6645	0.9998	9.1 × 10^−3^	76.2
Cloudy	No additives	y= −0.0068x + 0.6733	0.9755	6.8 × 10^−3^	101.9
	Ascorbic acid	y= −0.0142x + 0.6062	0.9537	14.2 × 10^−3^	48.8
	Cinnamon extract	y= −0.0085x + 0.6175	0.9912	8.5 ×10^−3^	81.5
	Clove extract	y= −0.0067x + 0.6043	0.9871	6.7 × 10^−3^	103.4
cyanidin-3-*O*-arabinoside					
Clarified	No additives	y = −0.0065x + 1.7102	0.9874	6.5 × 10^−3^	106.6
	Ascorbic acid	y = −0.0105x + 1.6880	0.9536	10.5 ×10^−3^	66.0
	Cinnamon extract	y = −0.0090x + 1.6972	0.9947	9.0 × 10^−3^	77.0
	Clove extract	y = −0.0088x + 1.7321	0.9996	8.8 × 10^−3^	78.8
Cloudy	No additives	y = −0.0066x + 1.7454	0.9953	6.6 × 10^−3^	105.0
	Ascorbic acid	y = −0.0124x + 1.6865	0.9635	12.4 × 10^−3^	55.9
	Cinnamon extract	y = −0.0084x + 1.6881	0.9930	8.4 × 10^−3^	82.5
	Clove extract	y = −0.0064x + 1.6794	0.9918	6.4 × 10^−3^	108.3
cyanidin-3-*O*-xyloside					
Clarified	No additives	y = −0.0076x + 0.9343	0.9343	7.6 × 10^−3^	91.2
	Ascorbic acid	y = −0.0116x + 0.8695	0.9430	11.6 × 10^−3^	59.7
	Cinnamon extract	y = −0.0098x + 0.8939	0.9947	9.8 × 10^−3^	70.7
	Clove extract	y = −0.0095x + 0.9329	1.000	9.5× 10^−3^	72.9
Cloudy	No additives	y = −0.0073x + 0.9528	0.9794	7.3× 10^−3^	94.9
	Ascorbic acid	y = −0.0139x + 0.9083	0.9593	13.9 × 10^−3^	49.9
	Cinnamon extract	y = −0.0093x + 0.8911	0.9929	9.3 × 10^−3^	74.5
	Clove extract	y = −0.0073x + 0.9010	0.9894	7.3 × 10^−3^	94.9

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
