# Peer review of "The Effect of Plant Additives on the Stability of Polyphenols in Cloudy and Clarified Juices from Black Chokeberry (Aronia melanocarpa)"

_antioxidants, 2020, doi:10.3390/antiox9090801_

Round 1
Reviewer 1 Report
The authors present the effect of using plant extracts as additives on the polyphenolic stability of Aronia melanocarpa juices.
I have several comments:
-in the introduction I think they would need to introduce information about the phenolic content of the spices they have chosen for this work, although it is specified later in section 3.1.
-in section 2.3, a chromatographic method with such a high pH (4.5% formic acid) is normal for the analysis of flavonols and anthocyanins, but not for the analysis of polyphenols. Can the authors reference this method? They should introduce (for example in section 3.1) some standard chromatograms for each wavelength
-in section 2.4 I think a table with the m/z ions of the identified compounds would be interesting. It is also necessary that they include the information corresponding to the calibration curves.
Author Response
Journal: Antioxidants
Title: The effect of plant additives on the stability of polyphenols in cloudy and clarified juices from black chokeberry (Aronia melanocarpa)
Manuscript ID: antioxidants-882476
Please find below detailed references to Reviewer’s comments.
Response to the Reviewer
Authors appreciate the comments and suggestions to present research report.
We have thoroughly revised the Manuscript according to reviewers’ comments and responded to each comment, point by point.
Reviewer #1:
The authors present the effect of using plant extracts as additives on the polyphenolic stability of Aronia melanocarpa juices.
I have several comments:
Comment 1. in the introduction I think they would need to introduce information about the phenolic content of the spices they have chosen for this work, although it is specified later in section 3.1.
Response: The sentence has been implemented in line 65-75: “Both spices, cinnamon and cloves contain polyphenolic compounds. Cinnamon contains phenolic acids (caffeic, chlorogenic, ferulic, p-coumaric, p-hydroxybenzoic, protocatechuic, rosmarinic, syringic), flavonols (rutin, quercetin, quercitrin, kaempferol, isorhamentin), flavone (apigenin), flavanols (catechin, epicatechin), procyanidins, and non-flavonoid phenoglycosides [18-21]. In contrast, cloves contain phenolic acids (gallic, caffeic), flavonols (tamarixetin 3-O-b-D-glucopyranoside, ombuin 3-O-b-D-glucopyranoside, quercetin) and ellagitannins (casuarictin, eugeniin, tellimagrandin I and 1,3-di-Ogalloyl-4,6 -O- [S] -hexahydroxydiphenoyl-b-D-glucose [22-24].”
Vallverdú-Queralt, A.; Regueiro, J.; Martínez-Huélamo, M.; Alvarenga, J. F.; Leal, L.N.; Lamuela-Raventos, R.M.A comprehensive study on the phenolic profile of widely used culinary herbs and spices: Rosemary, thyme, oregano, cinnamon, cumin and bay. Food Chemistry 2014, 154, 299-307.
Luo, Q.; Wang, S.; Lu, Q.; Luo, J.; Cheng, Y. Identification of compounds from the water soluble extract of cinnamomum cassia barks and their inhibitory effects against high-glucose-induced mesangial cells. Molecules 2013, 18(9), 10930-10943.
Al-Numair, K.S.; Ahmad, D.; Ahmed, S.B.; Al-Assaf, A.H. Nutritive value, levels of polyphenols and anti-nutritional factors in Sri Lankan cinnamon (Cinnamomum Zeyalnicum) and Chinese Cinnamon (Cinnamomum Cassia). Food Science & Agriculture Research Center, King Saud University 2007, 154, 5-21.
Kalvėnienė, Z.; Velžienė, S.; Ramanauskienė, K.; Savickas, A.; Ivanauskas, L.; Brusokas, V. The qualitative analysis of ethanol extracts of herbal raw materials by method of high pressure liquid chromatography. Acta Poloniae Pharmaceutica-Drug Research 2007, 64, 327-333.
Sanae, F.; Kamiyama, O.; Ikeda-Obatake, K.; Higashi, Y.; Asano, N.; Adachi, I.; Kato, A. Effects of eugenol-reduced clove extract on glycogen phosphorylase b and the development of diabetes in db/db mice. Food Function 2014, 5(2), 214-219.
Kim, J.; Seo, C.; Kim, S.; Ha, H. Simultaneous Determination of gallic acid, ellagic acid, and eugenol in Syzygium aromaticum and verification of chemical antagonistic effect by the combination with curcuma aromatica using regression analysis. Journal of Analytical Methods in Chemistry 2013, Article ID 375294, 1-7.
Rastogi, S.; Pandey, M.M.; Rawat, A.K. High-Performance Thin-Layer chromatography densitometric method for the simultaneous determination of three phenolic acids in Syzygium aromaticum (L.) Merr. & Perry. Journal of AOAC International 2008, 91(5), 1169-1173.
Comment 2. in section 2.3, a chromatographic method with such a high pH (4.5% formic acid) is normal for the analysis of flavonols and anthocyanins, but not for the analysis of polyphenols. Can the authors reference this method? They should introduce (for example in section 3.1) some standard chromatograms for each wavelength.
Response: The following citations has been added:
Ochmian, I.; Grajkowski, J.; Smolik, M. Comparsion of some morphological features, quality and chemical content of four cultivars of chokeberry fruits (Aronia melanocarpa). Notulae Botanicae Horti Agrobotanici Cluj-Napoca, 2012, 40, 253–260.
Ochmian, I.; Oszmiański, J.; Skupień, K. Chemical composition, phenolics and firmness of small black fruits. Journal of Applied Botany and Food Quality. 2009, 83, 64–69.
The paper does not present standard chromatograms for each wavelength, because these determinations were performed in a different laboratory and we did not have access to them, we only obtained the results in numerical form, which were included in the paper.
Comment 3. in section 2.4 I think a table with the m/z ions of the identified compounds would be interesting. It is also necessary that they include the information corresponding to the calibration curves.
Response: I agree with the Reviewer's suggestion that the table with the m / z ions of the identified compounds would be interesting and the information corresponding to the calibration curves should be included, however, the above-mentioned data were not included in the paper because they were not new methods for determining these compounds. Moreover, likewise above those determinations were performed in a different laboratory and we did not have access to specifications.
Reviewer 2 Report
Dear Authors,
I reviewed with pleasure the paper “The effect of plant additives on the stability of polyphenols in cloudy and clarified juices from black chokeberry (Aronia melanocarpa)”.
The study design is appropriate, and methods are sound.
However, I suggest arranging differently the section “results and discussion” in order to clearly report:
- the effect of cinnamon/clove extract addition on phenolic compounds in chokeberry juice;
- the effect of ascorbic acid addition on phenolic compounds in chokeberry juice;
- differences/similarities between clarified/cloudy juice;
- the effect of pasteurization.
The current organization of results is a bit confusing.
It might be interesting to report degradation kinetic curves.
Detailed comments are reported in the attached file.

Author Response
Journal: Antioxidants
Title: The effect of plant additives on the stability of polyphenols in cloudy and clarified juices from black chokeberry (Aronia melanocarpa)
Manuscript ID: antioxidants-882476
Please find below detailed references to Reviewer’s comments.
Response to the Reviewer
Authors appreciate the comments and suggestions to present research report.
We have thoroughly revised the Manuscript according to reviewers’ comments and responded to each comment, point by point.
Reviewer #2:
Comment: I reviewed with pleasure the paper “The effect of plant additives on the stability of polyphenols in cloudy and clarified juices from black chokeberry (Aronia melanocarpa)”.
The study design is appropriate, and methods are sound.
However, I suggest arranging differently the section “results and discussion” in order to clearly report:
- the effect of cinnamon/clove extract addition on phenolic compounds in chokeberry juice;
- the effect of ascorbic acid addition on phenolic compounds in chokeberry juice;
- differences/similarities between clarified/cloudy juice;
- the effect of pasteurization.
The current organization of results is a bit confusing.
It might be interesting to report degradation kinetic curves.
Detailed comments are reported in the attached file.
Response: Detailed responses are as follows:
The section “results and discussion” has been rearranged according to Reviewer’s suggestion (line 166-382).
Line 17: space has been removed.
Line 32: Please, add reference.
The reference had been added:
Borowska, S., Brzóska, M. M. . Chokeberries (Aronia melanocarpa) and Their Products as a Possible Means for the Prevention and Treatment of Noncommunicable Diseases and Unfavorable Health Effects Due to Exposure to Xenobiotics. Comprehensive Reviews in Food Science and Food Safety 2016, 15(6), 982-1017. doi:10.1111/1541-4337.12221
Line 38: Please, add a comma after "acids".
Comma has been added.
Line 39: The need of processing is due to the fruit short shelf-life. Please, amend the text.
The sentence has been corrected: “The availability of fresh fruits is periodic (from mid-August to late September), and the short shelf-life necessitates further processing.”
Line 43: Please, add a comma after "processing".
Comma has been added.
Line 62: Please, add a comma after "acid".
Comma has been added.
Line 70: The title is not comprehensive. In the paragraph you also describe the preparation of extracts from cinnamon and clove, and the preparation of juice samples. Please, amend the title.
The title has been corrected for: “2.2. Sample preparation”
Line 71: The sentence is not clear. Please, explain the meaning of "research material".
The sentence: “Black chokeberry fruits (Aronia melanocarpa), cloves and cinnamon were used as the research material.” has been removed as unnecessary.
Line 85: The sentence sounds as an addition of ascorbic acid to chokeberry juice added with cinnamon or clove extract. in contrast, you added ascorbic acid to chokeberry juice with no addition of spice extracts. Please, amend the text by making this point clearer.
The sentence: “Ascorbic acid was added to final concentration 300 mg/L according to nutrition standards [10].” has been corrected for: “Juices with ascorbic acid contains ascorbic acid addition to final concentration 300 mg/L according to nutrition standards [27].”
Line 91: The paragraph reports the analysis of phenolic compounds in chokeberry juice, not in chokeberries. Please, amend the title.
The title has been amended for: “2.3. HPLC analysis of chokeberry juices/drinks polyphenolic compounds”
Line 92-93: Why did you use this mixture for juice dilution? Please, explain or add reference(s).
The reference has been added: “Lachowicz, S.; Oszmiański, J.; Pluta, S. The composition of bioactive compounds and antioxidant activity of Saskatoon berry (Amelanchier alnifolia Nutt.) genotypes grown in central Poland. Food Chemistry 2017, 235, 234-243.”
Line 108: Did you mean "spice plant extracts"?
The title has been corrected for: “2.4. Analysis of polyphenolic compounds of spice plants extracts in the LC-MS system”
Line 151-159: It might be useful to add a table reporting the polyphenol profile of cinnamon and clove aqueous extracts.
Thank you for a comment. At the beginning of the work, we inserted suggested table, but due to the different number and type of evaluated compounds in the extracts, the table did not present the data in a satisfactory way, so we decided to include this data in the text.
Line 160: Table 1 reports i) data on clarified/cloudy juices, ii) data on pasteurised/unpausterised juiuces; iii) data on juices added with cinnamon/clove/ascorbic acis. I would expect a more appropriate discussion. Please, amend the text (lines 160-170) by highlighting differences/similarities of adding cinnamon/clove extracts or ascorbic acids. Moreover, discuss the effect of pasteurization on phenolic acids and flavanol content. Discuss differences/similarities between cloudy/clarified juices, as well.
The section “results and discussion” has been rearranged and supplemented according to Reviewer’s suggestion (line 193-472).
Line 167: "little" with respect to what?
Text has been corrected for: “The juices contained very little catechin and (-)-epicatechin, regardless of the additive used, in relation to the other compounds found in the tested juices / drinks.”
Line 169: You state that it (+)-catechin content is usually omitted. Hence, why did you determine it?
The sentence has been added: “It should be noted that their evaluation is advisable, because technological processes can affect the decomposition of proanthocyanidins, also known as condensed tannins, which building blocks include catechin and epicatechin [37].”
Rauf, A.; Imran, M.; Abu-Izneid, T.; Iahtisham-Ul-Haq, Patel, S.; Pan, X.; Naz, S.; Sanches Silva, A.; Saeed, F.; Rasul Suleria, H. A. Proanthocyanidins: a comprehensive review. Biomedicine and Pharmacotherapy 2019, 116, 108999.
Line 181 – Table 1: In my opinion the caption is not appropriate. You do not report just the "changes in the content of phenolic acids and flavanols during pasteurization...". You also report data of clarified/cloudy juices and juices added with cinnamon/clove/ascorbic acids. Please, amend the caption.
The caption has been amended to: “Phenolic acids and flavanols content before and after juices / drinks pasteurization.”
Line 184 – Table 2: Same comment as Table 1 caption.
The caption has been amended to: “Table 2. Flavonol content before and after juices / drinks pasteurization.”
Line 185 – Table 3: Same comment as Table 1 caption.
The caption has been amended to: “Table 3. Cyanidin content before and after juices / drinks pasteurization.”
Line 188-198 : This comment might be moved in paragraph 3.1.
We appreciate the comment, however in our opinion we would like to stay with the original work structure.The idea of inserting the above-mentioned description as a separate paragraph was to collect and summarize the results showing the impact of technological processes on the polyphenolic compounds tested, and in our opinion, moving this fragment to paragraph 3.1 will impoverish this fragment of the work.
Line 299 – Table 7: You might replace data in tables by degradation kinetic curves.
I agree with the Reviewer's opinion that replacing the data in the tables with degradation kinetic curves would be a better form of data presentation, however, we will ask for your understanding, but due to the pandemic and the vacation period, it is difficult to make all corrections in such a short time required by the journal.
Reviewer 3 Report
The authors focused their attention on the analyse of the influence of cinnamon or clove extracts addition on the concentration of selected polyphenols content in chokeberry juices, when this drinks undergo technological processes, such as clarifying and pasteurisation. This study appears quite interesting and the authors well conducted the research project, however some aspects should be well clarified before the acceptance of the paper.
In particular, the authors should well explain as the amount of cinnamon or clove extracts to add was chosen and if others experiments were performed.
In the introduction section the authors should better explain the novelty of this research.
The English should be revised in the whole manuscript.
In my opinion this manuscript should be accepted only after these modifications..
Author Response
Journal: Antioxidants
Title: The effect of plant additives on the stability of polyphenols in cloudy and clarified juices from black chokeberry (Aronia melanocarpa)
Manuscript ID: antioxidants-882476
Please find below detailed references to Reviewer’s comments.
Response to the Reviewer
Authors appreciate the comments and suggestions to present research report.
We have thoroughly revised the Manuscript according to reviewers’ comments and responded to each comment, point by point.
Reviewer #3:
Comment. The authors focused their attention on the analyze of the influence of cinnamon or clove extracts addition on the concentration of selected polyphenols content in chokeberry juices, when this drinks undergo technological processes, such as clarifying and pasteurization. This study appears quite interesting and the authors well conducted the research project, however some aspects should be well clarified before the acceptance of the paper. In particular, the authors should well explain as the amount of cinnamon or clove extracts to add was chosen and if others experiments were performed.
In the introduction section the authors should better explain the novelty of this research. The English should be revised in the whole manuscript.
In my opinion this manuscript should be accepted only after these modifications..
Response: Detailed responses are as follows:
In particular, the authors should well explain as the amount of cinnamon or clove extracts to add was chosen and if others experiments were performed.
Line 95-96- the explanation is included in the following part: “… while the addition of extracts at the level of 5% was preceded by sensory optimization using a consumer panel.” (detailed data are not shown, since it is a subject of other publication in progress).
In the introduction section the authors should better explain the novelty of this research.
The sentence has been added: “In this study, the profile of bioactive compounds in chokeberry juices enriched with cinnamon or clove extracts subjected to processing has been analysed for the first time. Therefore the aim of the study was to analyze the influence of spices extracts addition, the technological processes (clarifying, pasteurization) and the storage time on the changes of selected polyphenols content in chokeberry juices.”
The English should be revised in the whole manuscript.
The English of the manuscript has been thoroughly revised.
Reviewer 4 Report
Overview
This paper is about the impact that spice and antioxidant addition has on the retention of polyphenolic contents in cokeberry drinks prepared with and without pasteurisation and filtration. It is based on the hypothesis that added antioxidants could help prevent degradation of natural polyphenolic antioxidants from the juices during storage.
The research follows a good scientific method and is well-presented statistically, although there are extensive tables on changes in content as a function of process and storage variables. This data could be assigned to on-line data with the paper itself presenting illustrative figures and the equations for degradation.
The scientific novelty for the juice processing area generally is low as most of these types of effects found have been published elsewhere and some key areas of mechanistic interpretation could be improved . However it is still of scientific interest for processors developing juice-herbal mixtures with improved shelf stability.
Specific comments:
Abstract:
“The plant has numerous advantages, however the most interesting is great health-promoting and antioxidative potential” This sentence is informal sales promotion and should be replaced with a technical sentence on why reducing the loss of polyphenols in the drink is required.
“The research showed that, despite the common view about the beneficial effects of polyphenols and other compounds exhibiting antioxidative potential to each other” Needs improved clarity: it is not a common view but an assumption without much foundation .
L34 This is a self-reference that is about polyphenolic antioxidant content, not about yield , cultivation or health effects. Better primary source references are available.
L38 This does review some of the potential health impacts but there are primary literature sources of the content that should be cited and credited rather than just a review
L44 This sweeping statement “a widely accepted view that the addition of antioxidants from other sources may prevent the loss of native compounds” is a key reason for the paper but is not supported by any argument or reference. It is better described as a misconception but the authors should give a better justification argument and evidence for the statement.
L80 detail on filtering and its effectiveness on clarity is not given
L83-84 there was 100 mL in 120 mL jars so a 20% air gap. State what jar material (glass/plastic) and closures were used and if needed how oxygen impermeable they were
L 86 Nutritional standards for Vit c are set for the end of storage life not the start. Its practice to use up to 4-6 fold excess of Vit C at the start of storage to compensate for the oxygen losses through thin walled plastic containers during shelf life at room temperatures
L88 Non pasteurised juices were stored at room temperature. Details of microbial stability are needed to interpret the subsequent data.
L188-213 The losses of anthocyanins are discussed without consideration of the assay technology. HPLC records loss when dimer formation occurs on heating although this can partly reverse on aging. Total anthocyanins by pH difference color measurement records all anthocyanins including dimers and complexes. Please check that comparison of figures on losses is not influenced by different assay methods.
254 Review to check that spice plant extract degradation was not potentially by complexation not picked up by HPLC assay. The health effects may still be present
347-349 These 3 sentences are not conclusions of the research which should be focused don new findings .
Author Response
Journal: Antioxidants
Title: The effect of plant additives on the stability of polyphenols in cloudy and clarified juices from black chokeberry (Aronia melanocarpa)
Manuscript ID: antioxidants-882476
Please find below detailed references to Reviewer’s comments.
Response to the Reviewer
Authors appreciate the comments and suggestions to present research report.
We have thoroughly revised the Manuscript according to reviewers’ comments and responded to each comment, point by point.
Reviewer #4:
Comment: This paper is about the impact that spice and antioxidant addition has on the retention of polyphenolic contents in chokeberry drinks prepared with and without pasteurisation and filtration. It is based on the hypothesis that added antioxidants could help prevent degradation of natural polyphenolic antioxidants from the juices during storage.
The research follows a good scientific method and is well-presented statistically, although there are extensive tables on changes in content as a function of process and storage variables. This data could be assigned to on-line data with the paper itself presenting illustrative figures and the equations for degradation.
The scientific novelty for the juice processing area generally is low as most of these types of effects found have been published elsewhere and some key areas of mechanistic interpretation could be improved. However it is still of scientific interest for processors developing juice-herbal mixtures with improved shelf stability.
Specific comments:
Abstract:
“The plant has numerous advantages, however the most interesting is great health-promoting and antioxidative potential” This sentence is informal sales promotion and should be replaced with a technical sentence on why reducing the loss of polyphenols in the drink is required.
Line 18-20: The following sentence has been added: “Recent research has strengthened the position of chokeberry as a source of phenolic compounds, antioxidants with high pro-health values, therefore it is important to investigate other substances protecting biologically active compounds during juice processing.”
“The research showed that, despite the common view about the beneficial effects of polyphenols and other compounds exhibiting antioxidative potential to each other” Needs improved clarity: it is not a common view but an assumption without much foundation.
Line 26-28: the sentence has been corrected for: “The research showed that, despite the common view about the beneficial effects of polyphenols and other compounds exhibiting mutual antioxidative potential, it is very difficult to inhibit the degradation process.”
The meaning of this sentence was to point that despite the high content and activity of polyphenolic compounds contained in the components, the activity in a complex system, i.e. a mixture, cannot be predicted. In the case of our research, it was found that these compounds did not have a strengthening but antagonistic effect in the mixture subjected to processing.
L34 This is a self-reference that is about polyphenolic antioxidant content, not about yield , cultivation or health effects. Better primary source references are available.
Literature cited has been changed for:
Kawecki, Z., Tomaszewska, Z. The effect of various soil management techniques on growth and yield in the black chokeberry (Aronia melanocarpa Elliot). Journal of Fruit and Ornamental Plant Research 2006, 14, 67-73
Strik, B., Finn, C., Wrolstad, R. Performance Of Chokeberry (Aronia Melanocarpa) In Oregon, Usa. Acta Horticulturae 2003, (626), 439-443.
L38 This does review some of the potential health impacts but there are primary literature sources of the content that should be cited and credited rather than just a review
Initially we have decided to cite the review in the described subject of the research since we were obliged to limit the references in the manuscript. In our opinion citation of the review papers should be acceptable since the data contained therein direct the reader to selected and most significant works in a given subject area. However according to Reviewer’s suggestion we incorporated the following citations:
Broncel M., Kozirog M., Duchnowicz P., Koter-Michalak M., Sikora J., Chojnowska-Jezierska J. Aronia melanocarpa extract reduces blood pressure, serum endothelin, lipid, and oxidative stress marker levels in patients with metabolic syndrome. Medical Science Monitor 2010, 16, 28–34
Kardum, N., Milovanović, B., Šavikin, K., Zdunić, G., Mutavdžin, S., Gligorijević, T., Spasić, S. Beneficial Effects of Polyphenol-Rich Chokeberry Juice Consumption on Blood Pressure Level and Lipid Status in Hypertensive Subjects. Journal of Medicinal Food 2015 18(11), 1231-1238.
Naruszewicz, M., Łaniewska, I., Millo, B., Dłużniewski, M. Combination therapy of statin with flavonoids rich extract from chokeberry fruits enhanced reduction in cardiovascular risk markers in patients after myocardial infraction (MI). Atherosclerosis 2007, 194(2), 179-184.
Poreba, R., Skoczynska, A., Gac, P., Poreba, M., Jedrychowska, I., Affelska-Jercha, A., Turczyn, B., Wojakowska, A., Oszmianski, J., Andrzejak, R. Drinking of chokeberry juice from the ecological farm Dzieciolowo and distensibility of brachial artery in men with mild hypercholesterolemia. Annals of Agricultural and Environmental Medicine 2009, 16(2), 305-308.
Sikora, J., Broncel, M., Markowicz, M., Chałubiński, M., Wojdan, K., Mikiciuk-Olasik, E. Short-term supplementation with Aronia melanocarpa extract improves platelet aggregation, clotting, and fibrinolysis in patients with metabolic syndrome. European Journal of Nutrition 2011, 51(5), 549-556.
Sikora, J., Broncel, M., Mikiciuk-Olasik, E. Aronia melanocarpa Elliot Reduces the Activity of Angiotensin I-Converting Enzyme—In Vitro and Ex Vivo Studies. Oxidative Medicine and Cellular Longevity 2014, Article ID 739721, 1-7.
L44 This sweeping statement “a widely accepted view that the addition of antioxidants from other sources may prevent the loss of native compounds” is a key reason for the paper but is not supported by any argument or reference. It is better described as a misconception but the authors should give a better justification argument and evidence for the statement.
The sentence has been removed and corrected for: “The results of many studies do not support the fact that the addition of antioxidants from other sources prevents loss of native compounds. In fact, the unpredictable effect of enhancing or suppressing activity in a mixture of different active compounds is often overlooked.”
L80 detail on filtering and its effectiveness on clarity is not given
The sentence has been corrected: “Part of the juice was clarified by membrane filtering. The efficiency of the process was visually determined by the absence of suspended high molecular solids such as protein. “
L83-84 there was 100 mL in 120 mL jars so a 20% air gap. State what jar material (glass/plastic) and closures were used and if needed how oxygen impermeable they were
The sentence has been corrected for: “Juices were poured into a glass jars with galvanized and varnished steel lid with a gasket closures, which after closing and pasteurization process were secured against oxygen permeability.”
L 86 Nutritional standards for Vit C are set for the end of storage life not the start. Its practice to use up to 4-6 fold excess of Vit C at the start of storage to compensate for the oxygen losses through thin walled plastic containers during shelf life at room temperatures.
We agree and appreciate the suggestion, however, the study aimed to test the potential effect of vitamin C on the stability of polyphenols in tested juices / drinks. Prior storage the samples were protected against oxygen by the use of glass jars with a sealed lid.
L88 Non pasteurized juices were stored at room temperature. Details of microbial stability are needed to interpret the subsequent data.
Unpasteurized juices / drinks were not stored, such a process was conducted on samples without additives subjected to pasteurization, therefore no microbial stability has been required.
L188-213 The losses of anthocyanins are discussed without consideration of the assay technology. HPLC records loss when dimer formation occurs on heating although this can partly reverse on aging. Total anthocyanins by pH difference color measurement records all anthocyanins including dimers and complexes. Please check that comparison of figures on losses is not influenced by different assay methods.
The following sentence has been added: “The anthocyanin content is sometimes higher in the differential method from HPLC when the results are expressed as equivalents of the same compound. If anthocyanin standards are used in HPLC, the sum of anthocyanins is rather higher than the result obtained by the differential method [45]. In 6 out of 7 juices / drinks using the differential method performed in cuvettes or plates, the content of anthocyanins was lower than the results of measurements from two HPLC systems [46].”
Lee, S.G.; Vance, T.M.; Nam, T.; Kim, D.; Koo, S.I.; Chun, O.K. Evaluation of pH differential and HPLC methods expressed as cyanidin-3-glucoside equivalent for measuring the total anthocyanin contents of berries. Journal of Food Measurement and Characterization 2016, 10(3), 562-568
Lee, J.; Rennaker, C.; Wrolstad, R.E. Correlation of two anthocyanin quantification methods: HPLC and spectrophotometric methods. Food Chemistry 2008, 110(3), 782-786.
254 Review to check that spice plant extract degradation was not potentially by complexation not picked up by HPLC assay. The health effects may still be present.
The comment is not clear for us. We reviewed to check that spice plant extract degradation was not potentially by complexation not picked up by HPLC assay. It is possible that there are constant changes in the structures of compounds, that can combine with each other and then break down again into smaller particles. Therefore it is likely to occur and the health effects may still be present.
347-349 These 3 sentences are not conclusions of the research which should be focused don new findings .
The sentences has been changed for: “Fresh chokeberry juice has high polyphenol content, which is related to potential health-promoting effect, however processing and storage could often influence polyphenols losses.”
Round 2
Reviewer 1 Report
From my point of view, in this new version, the manuscript and in particular the introduction has improved considerably. It is a problem that do not have chromatograms to accompany the text. anyway creoq ue the manuscript is acceptable to publish
Reviewer 2 Report
The paper has been amended as suggested.